# Integration of epigenetic and genetic profiles identifies multiple sclerosis disease-critical cell types and genes

Qin Ma [1,3], Hengameh Shams[1,3], Alessandro Didonna[2], Sergio E. Baranzini [1], Bruce A. C. Cree[1], Stephen L. Hauser [1], Roland G. Henry [1] & Jorge R. Oksenberg [1✉]

Genome-wide association studies (GWAS) successfully identified multiple sclerosis (MS) susceptibility variants. Despite this notable progress, understanding the biological context of these associations remains challenging, due in part to the complexity of linking GWAS results to causative genes and cell types. Here, we aimed to address this gap by integrating GWAS data with single-cell and bulk chromatin accessibility data and histone modification profiles from immune and nervous systems. MS-GWAS associations are significantly enriched in regulatory regions of microglia and peripheral immune cell subtypes, especially B cells and monocytes. Cell-specific polygenic risk scores were developed to examine the cumulative impact of the susceptibility genes on MS risk and clinical phenotypes, showing significant associations with risk and brain white matter volume. The findings reveal enrichment of GWAS signals in B cell and monocyte/microglial cell-types, consistent with the known pathology and presumed targets of effective MS therapeutics.

[1] Weill Institute for Neurosciences, Department of Neurology, University of California San Francisco, San Francisco, CA 94158, USA. [2] Department of Anatomy and Cell Biology, Brody School of Medicine, East Carolina University, Greenville, NC 27834, USA. [3]These authors contributed Equally: Qin Ma, Hengameh Shams. ✉email: jorge.oksenberg@ucsf.edu

Multiple sclerosis (MS) is an autoimmune disease affecting the central nervous system (CNS), and common cause of non-traumatic neurological disability in young adults. MS pathogenesis is complex and multifactorial with a well-established, albeit partially understood polygenic susceptibility component. Genome-wide association studies (GWAS) have identified 201 independent genome-wide significant associations outside the major histocompatibility complex (*MHC*) and 32 within the *MHC* region, leading to a catalog of 551 candidate risk genes[1]. These variants, together with additional 416 suggestive effects, explain approximately half of the disease heritability. Like other complex diseases and traits, most MS-associated variants identified by GWAS map to noncoding segments of the genome and are concentrated in regulatory regions[2], thus likely contributing to risk through cell-type specific transcriptional regulatory mechanisms mediated by short- and long-range chromatin interactions. Using open epigenetic databases, the enrichment of disease variants and associated pathways operating primarily on immunological competent cells has been consistently reported, confirming the autoimmune model of pathogenesis[2–4]. More recent analyses of genomic data have also implicated microglia in driving disease risk[1]. Annotating the precise susceptibility gene roster and cellular compartments from GWAS datasets remain a priority and a key step for any translational application of genetic discoveries.

Here, we used the GARFIELD[5] classification approach on the latest MS GWAS dataset to perform a detailed enrichment analysis of regulatory annotations and describe the cellular basis of disease susceptibility. We further applied Hi-C-coupled multimarker analysis of genomic annotation (H-MAGMA)[6] to integrate GWAS and three-dimensional (3D) chromatin interaction profiles, and this approach led to the identification of cell-specific susceptibility genes in B cells, monocytes, and microglia. Finally, we leveraged the resulting information to develop cell-specific polygenic risk-scores (CPRS) to associate cell-specific genetics to clinical phenotypes of interest.

## Results

### MS GWAS-associated loci are enriched in open chromatin regions in microglia and peripheral immune cells.
To link MS susceptibility variants to genes active in specific cellular compartments, we integrated GWAS data with single-cell and bulk chromatin accessibility data, histone modification profiles, and 3D chromatin contacts information, following the workflow shown in Supplementary Fig. 1. Two studies have generated accessible chromatin reference maps from single-cell ATAC-seq (scATAC-seq) screenings on healthy peripheral blood and brain tissue from cognitively healthy individuals[7,8]. Building on the granularity of these datasets, the GARFIELD algorithm was applied to estimate the enrichment of MS GWAS associations in 6 primary brain cell types and 14 peripheral immune cell types. The distinct advantage of GARFIELD is that it accounts for major sources of confounding, which include minor allele frequency, distance to the nearest transcription start site, and the number of LD proxies ($r^2 > 0.8$). Significant enrichment of associated loci was observed in all peripheral immune cell types at four different GWAS *P*-value thresholds (T < $10^{-5}$ to T < $10^{-8}$), with slightly higher levels in naïve B cells (Fig. 1a). Within the brain cell types, GWAS signals were significantly enriched in microglia, but not in astrocytes, oligodendrocyte precursor cells (OPCs), oligodendrocytes, or neurons (Fig. 1a).

We next sought to extend the analysis using the algorithm default regulatory annotation of open chromatin regions (OCRs) at eight GWAS *P*-value thresholds (T < 0.05 to T < $10^{-8}$) in 424 cell lines or primary cell types. Blood was the most significantly enriched tissue type in MS risk genetics (Fig. 1b). Follow-up analysis of regulatory annotations denoting OCRs in the immune system and CNS available from the Encyclopedia of DNA Elements (ENCODE) and the Blueprint projects also found significant enrichments of GWAS signals in multiple immune cell types, with the highest levels of enrichment observed in B cells and monocytes from ENCODE and Blueprint data, respectively (Fig. 1c and Supplementary Fig. 2). Interestingly, modest enrichment of GWAS associations (odds ratio values range from 1.34 to 2.52 at GWAS *P*-value threshold T < $10^{-5}$) in the CNS chromatin accessibility datasets, including brain microvascular endothelial cells, was detected in this analysis (Fig. 1c).

Growing evidence suggests that MS has significant genetic correlations with autoimmune and neuropsychiatric disorders[6,9–11]. We tested therefore the cell-type-specific OCRs enrichment within 14 peripheral immune cell types and 6 brain cell types for GWAS associations in the context of systemic lupus erythematosus (SLE)[12], rheumatoid arthritis (RA)[13], celiac disease (CD)[14], inflammatory bowel disease (IBD)[15], systemic sclerosis (SS)[16], type 1 diabetes (T1D)[17], Alzheimer disease (AD)[18], schizophrenia (SCZ)[19], and bipolar disorder (BPD)[20]. As expected, we found the immune cell types enriched in MS also enriched in other autoimmune diseases, in particular RA, IBD and SLE, yet B cells and monocytes showed highest enrichment in MS (Supplementary Fig. 3). Noteworthy, a certain degree of enrichment for immune cells was also detected in the canonical neurological disease AD, BDP, and SCZ. Regulatory T cells, cytotoxic CD8+ T cells and memory CD8+ T cells have the highest level of enrichment in T1D, RA, and SLE respectively. The enrichment of microglia seems to genetically link MS, AD, and surprisingly, SLE.

### MS GWAS signals are enriched in active enhancer regions.
To gain additional insights into the regulatory function of genetic variants in immune cells, we examined the enrichment for MS GWAS risk loci in chromatin immunoprecipitation sequencing (ChIP-seq) peaks targeting key histone modifications (H3K27ac, H3K27me3, H3K36me3, H3K4me1, H3K4me3, and H3K9me3) available from the ENCODE and Blueprint projects. Significant overlaps between GWAS signals and H3K27ac, H3K4me3, and H3K4me1 ChIP-seq peak regions were observed (Fig. 2a and Supplementary Fig. 4), whereas B cells consistently represented the cytotype displaying the highest enrichment in genetic signals (Fig. 2a and Supplementary Fig. 4). H3K4me1 is enriched at active and primed enhancers, while H3K27ac is a marker for active enhancers and H3K4me3 is highly enriched at active promoters[21], suggesting that MS genetic risk associations are enriched at active enhancers and promoters.

Using the ENCODE Encyclopedia Registry of candidate cis-Regulatory Elements (cCREs), GWAS hits mapped to proximal and distal enhancer-like signatures (pELS and dELS), especially in B cells and monocytes (Fig. 2b). Altogether, the data indicate that MS-associated variants concentrate in active regulatory regions, especially in enhancer elements, and B cells and monocytes represent the main target cytotypes.

### Integration of genetic and 3D chromatin interaction data identifies putative causal genes.
Next, we applied the H-MAGMA framework to update the roster of cell-specific susceptibility genes, integrating the reported GWAS summary statistics (14,802 subjects with MS and 26,703 controls)[1] with promoter capture Hi-C[22] (PCHiC) and H3K4me3 HiChIP[23] datasets obtained from B cells, monocytes, and microglia, which are the most significantly impacted cellular compartments described above. Through this analysis, we identified 1247 genes in B cells, 1148 genes in monocytes, and 1183 genes in microglia

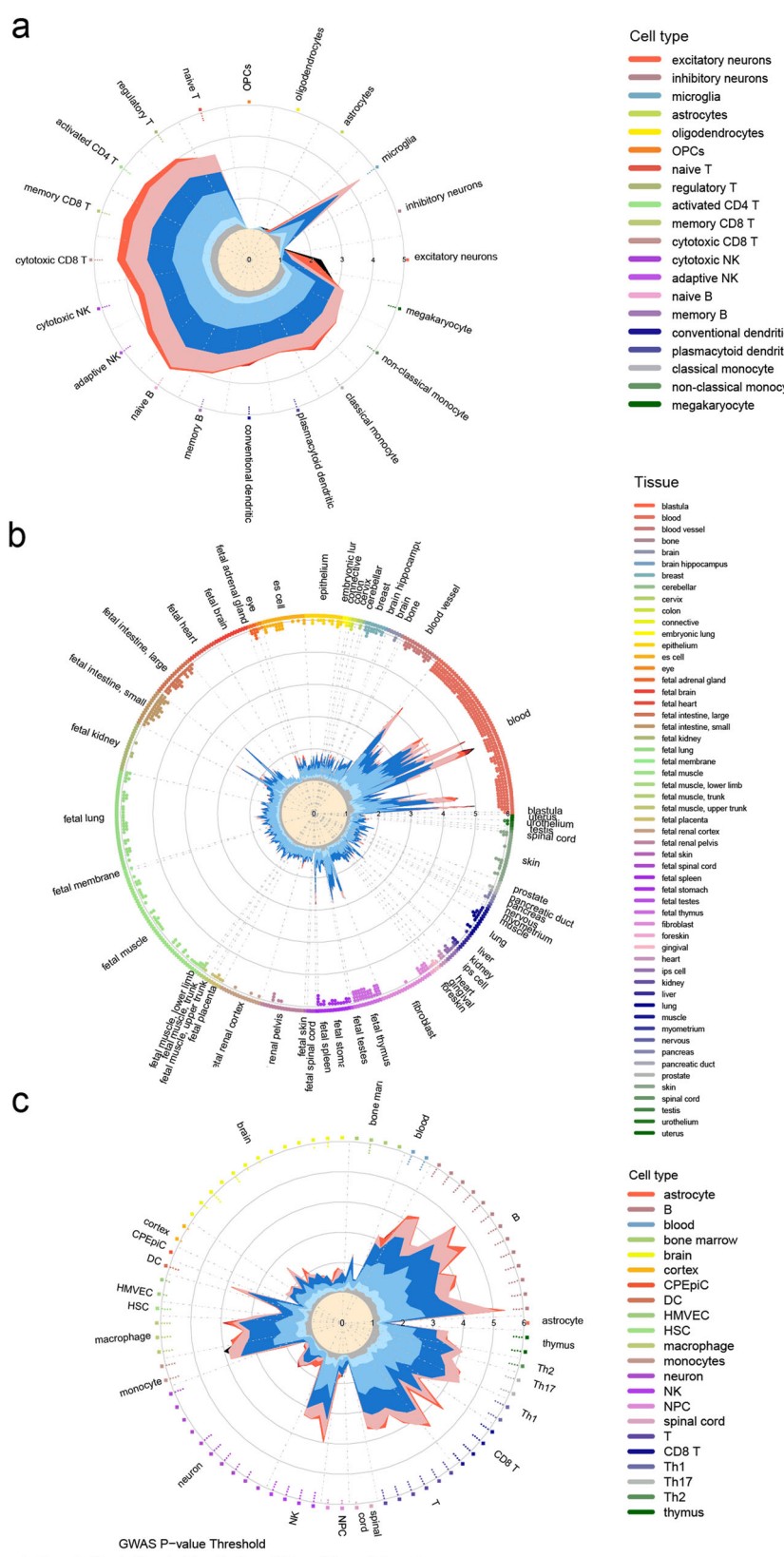

(FDR < 0.05) (Supplementary Data 1). A total of 717 genes are shared by all cell types, while 234, 136 and 281 genes are unique to B cells, monocytes, and microglia, respectively (Fig. 3a and Supplementary Data 2). Moreover, the cell-specific genes overlap with 283 (51.4%), 265 (48.1%), and 294 (53.4%) of the 551

previously prioritized genes based on cis expression quantitative trait loci effect (cis-eQTL) and regulatory networks[1] (Supplementary Data 3).

Gene ontology (GO) analysis on the 717 common genes highlighted an enrichment in immune-related pathways, with

**Fig. 1 Enrichment of MS GWAS associations in open chromatin regions (OCRs). a** MS GWAS enrichment at cell-type-specific OCRs within 6 primary brain cell types and 14 peripheral immune cell types derived from scATAC-seq. Radial lines show odds ratio (OR) values at eight GWAS *P*-value thresholds (T) for cell-type-specific ATAC-seq peaks from brain and immune cell types. **b** Radial lines show OR values at eight GWAS *P*-value thresholds (T) for 424 cell lines or primary cell types available from the GARFIELD software. **c** Radial lines show OR values at eight GWAS *P*-value thresholds (T) for immune system and CNS available from the ENCODE project. Dots in the inner ring of the outer circle denote significant GARFIELD enrichment (if present) at T < $10^{-5}$ (outermost) to T < $10^{-8}$ (innermost) after multiple-testing correction. The colors represent cell or tissue types. CPEpiC choroid plexus epithelial cell, DC dendritic cell, HMVEC brain microvascular endothelial cell, HSC hematopoietic stem cell, NK natural killer, NPC neural progenitor cell.

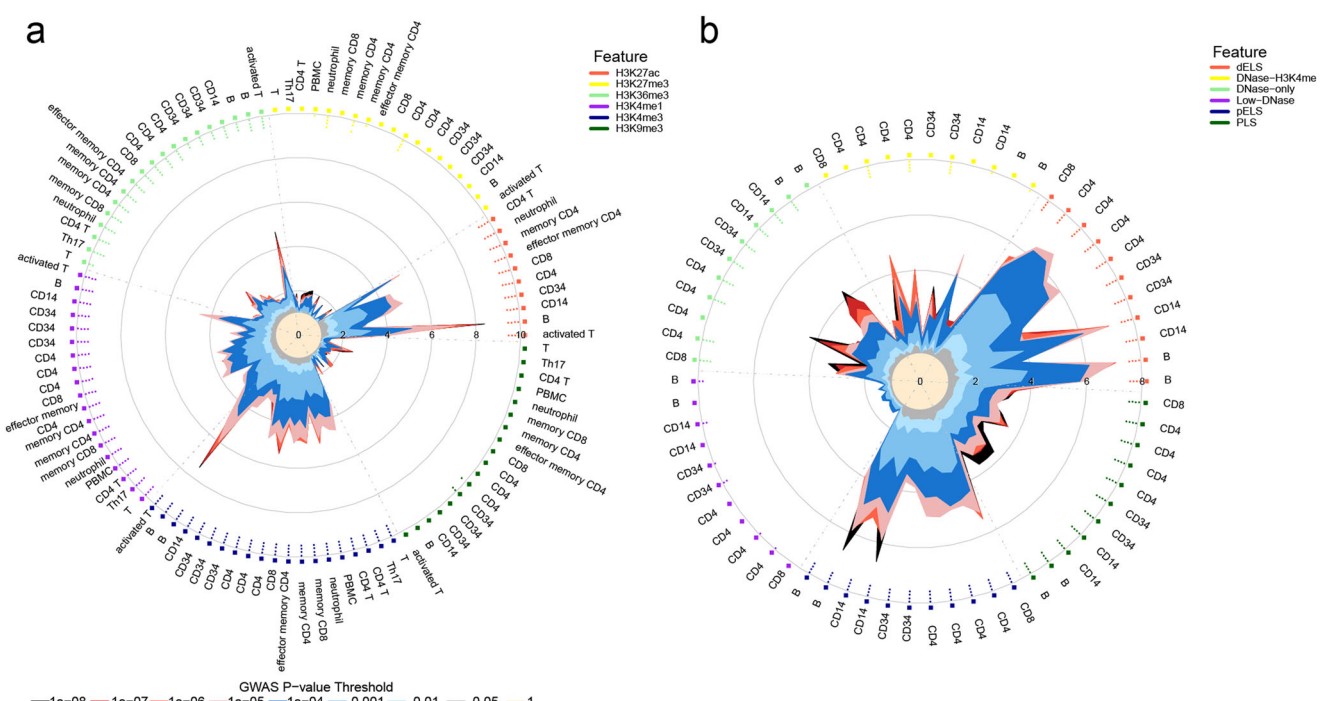

**Fig. 2 Enrichment of MS GWAS associations in histone modification ChIP-seq peaks. a** Radial lines show OR values at eight GWAS *P*-value thresholds (T) for histone modification peaks of immune cell types available from the ENCODE project. Dots in the inner ring of the outer circle denote significant GARFIELD enrichment (if present) at T < $10^{-5}$ (outermost) to T < $10^{-8}$ (innermost) after multiple-testing correction. The colors represent six kinds of histone modifications: H3K27ac, H3K27me3, H3K36me3, H3K4me1, H3K4me3, and H3K9me3. MS shows predominant enrichment in H3K27ac, H3K4me1, and H3K4me3. **b** Radial lines show OR values at eight GWAS *P*-value thresholds (T) for cCREs of immune cell types available from the ENCODE project. The colors represent different kinds of cCREs. dELS, distal-enhancer-like signatures, which have high DNase and H3K27ac signals and are not within 2 kb of an annotated transcription start site (TSS). DNase-only, cCREs with high DNase signals but low H3K4me3 and H3K27ac signals. Low-DNase, cCREs with low DNase signals in particular cell types. pELS, proximal-enhancer-like signatures which are within 2 kb of an annotated TSS and have high DNase and H3K27ac signals and have a low H3K4me3 signals if they are within 200 bp of an annotated TSS. PLS, promoter-like signatures which fall within 200 bp of an annotated GENCODE TSS and have high DNase and H3K4me3 signals.

"cytokine signaling in immune system" being the most significant term (Fig. 3b and Supplementary Data 4). GO analysis on the unique gene lists found "positive T cell selection", "neutrophil degranulation", and "PRC2 methylates histones and DNA" as the most significantly enriched categories for B cell, monocyte, and microglia, respectively (Fig. 3c and Supplementary Data 5). Notably, *DNMT3A*, which encodes the DNA (cytosine-5)-methyltransferase 3 A enzyme, is among the unique genes in microglia and drives the enrichment in the "PRC2 methylates histones and DNA" pathway. Interestingly, some of the MS associated SNPs (T < $10^{-5}$) are within the enhancer regions and/or interact with the promoter regions of *DNMT3A* in microglia (Supplementary Fig. 5), which further validates the possible functional effects of the SNPs.

**Cell-specific polygenic risk score analysis.** To link genetics to MS phenotypes in a cell-specific context, we generated polygenic risk scores (CPRS) based on B cell, monocyte, and microglia variants as well as a combined score comprising all the above-

mentioned cell types. The CPRS were computed for the white population in the UK Biobank and UCSF-EPIC datasets, incorporating the estimated variant effect sizes from the largest MS-GWAS study[1] to date. Prediction accuracies of best-performing CPRS in the validation set, UK Biobank phase 1 (UKBB1, 601 cases/109,990 controls), were examined in two independent test sets, UK Biobank phase 2 (UKBB2, 1354 cases/252,065 controls) and UCSF-EPIC (494 cases/449 controls), upon adjustment for genetic background assessed by the first twenty principal components of ancestry (Supplementary Table 1).

The predictive power of CPRS for MS risk is very similar across different cell subtypes as well as the combined score in all datasets, indicated by the coefficient of determination ($R^2$) corrected for disease prevalence in the white population in Western Europe[24] (0.00127) as well as the area under the curve (AUC) as shown in Supplementary Table 1. This observation is not surprising given the high degree of overlapping variants across the three cell-types (Supplementary Data 6) identified by the H-MAGMA algorithm. All CPRS are significantly associated with risk (*P* < 1e−20). Both significance and association levels are

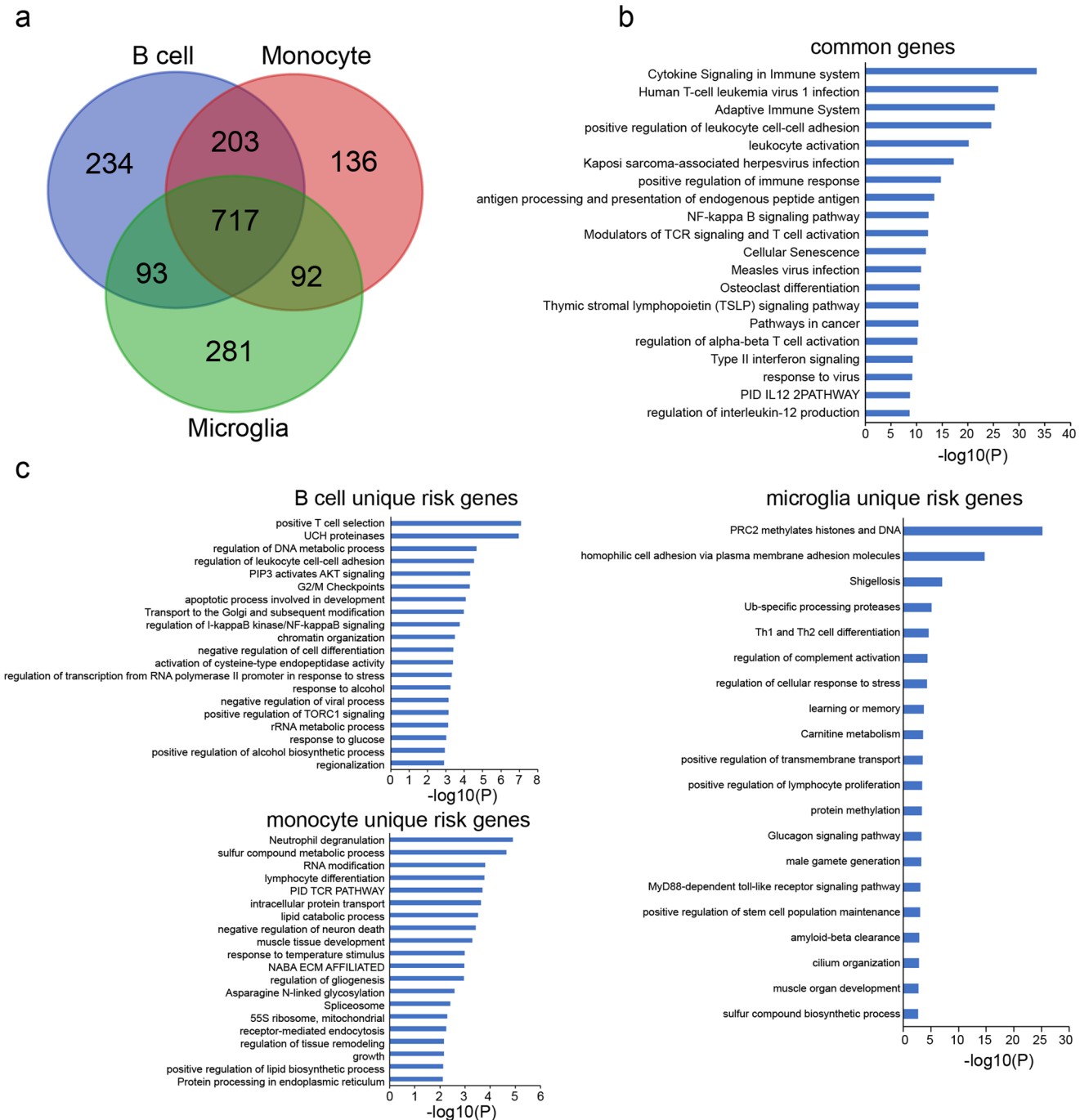

**Fig. 3 Functional enrichment of the common and unique genes. a** Venn diagrams showing the overlap of risk genes between each cell type. **b** Histograms showing the top 20 significant enriched pathways of the shared genes. **c** Histograms showing the significant enriched pathways of unique risk genes.

enhanced in UKBB2 compared to UKBB1. Such difference may be driven by self-reported disease status, likely leading to various percentages of false negatives across the two stages of data collection. Replicating risk association in the UCSF-EPIC cohort consisting of 494 neurologist-diagnosed MS cases and 449 controls resulted in notably increased $R^2$ and AUC compared to UK Biobank, due to both accurately recorded disease status and a higher case/control ratio. It should also be noted that EPIC subjects constituted a small portion of the International Multiple Sclerosis Genetics Consortium (IMSGC) discovery cohort, thereby, the effect sizes were readjusted upon exclusion of the EPIC dataset prior to risk score calculations in this cohort to avoid inflation in classification accuracy of polygenic scores.

Next, we examined the discriminative power of CPRS without the *MHC* region. The UKBB2 results expectedly showed a reduced predictive power of non-*MHC* CPRS according to all measures shown in Supplementary Table 2. Specifically, excluding the *MHC* burden from the cumulative CPRS decreased the AUC values shown in Supplementary Table 1 by 3% to 5% across all cell types. Due to the small cohort size and larger confidence intervals, the effect of *MHC* on CPRS accuracy in EPIC was less pronounced. Furthermore, CPRS was computed with unique variants listed in Supplementary Table 3 for each cell type, which resulted in most significant risk associations with monocytes ($R^2 = 2.4\%$, $P = 1.7e{-}84$) and B cells ($R^2 = 2.2\%$, $P = 1e{-}69$). These were also replicated in the EPIC dataset for both

| Table 1 Prediction accuracy of CPRS based on unique SNPs in each cell type. | | | | | | |
|---|---|---|---|---|---|---|
| **Model ($r^2 = 0.1$)** | **UKBB2 1354 cases/ 252,065 controls** | | | **UCSF-EPIC 494 cases/ 449 controls** | | |
| | **$R^{2*}$ (%)** | **P** | **AUC (%)** | **$R^{2*}$ (%)** | **P** | **AUC (%)** |
| B cell | 2.2 | 1e-69 | 64 | 2 | 2.3e-11 | 65 |
| Monocyte | 2.4 | 1.68e-84 | 64 | 3.1 | 4e-16 | 68 |
| Microglia | 2 | 7.2e-72 | 63 | 2 | 4e-11 | 65 |
| Combined | 2.9 | 2.9e-102 | 65 | 3.4 | 2e-17 | 69 |
| *Adjusted for MS prevalence 0.00127. | | | | | | |

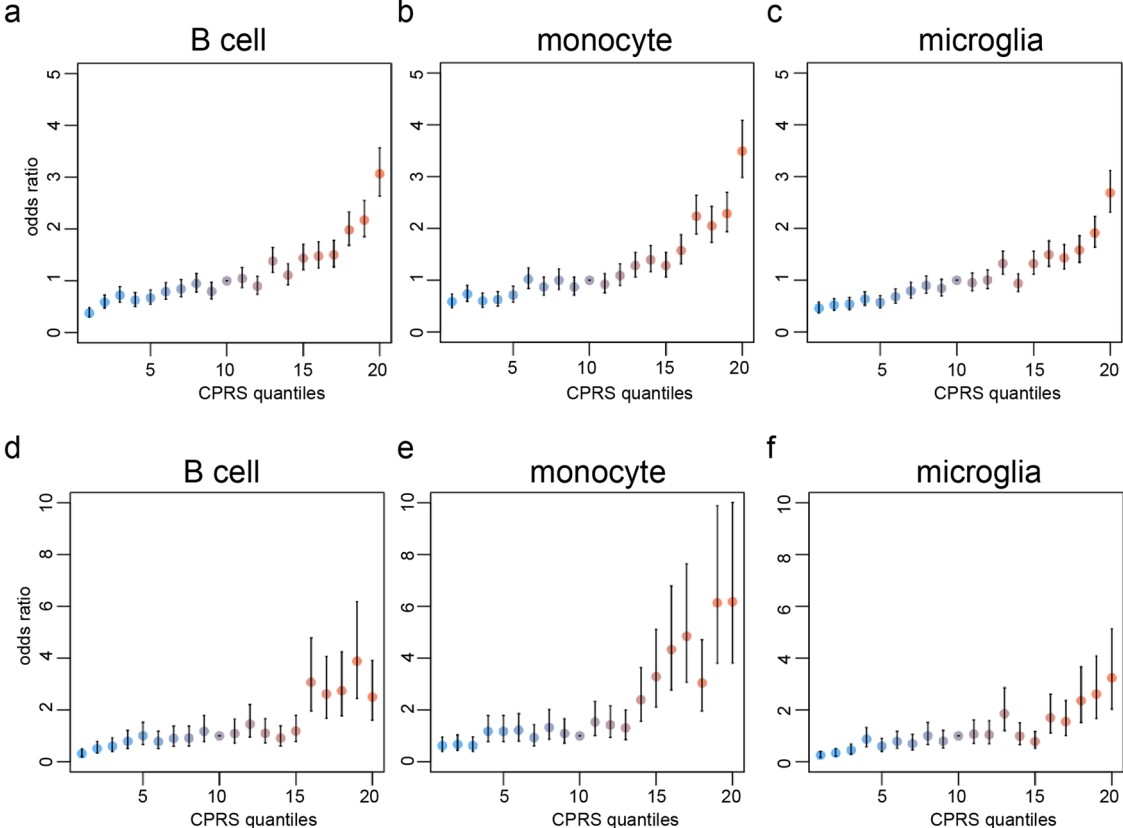

**Fig. 4 Odds ratio across CPRS strata B cell, monocyte, and microglia. a–c** UKBB2 and (**d–f**) UCSF-EPIC datasets. Cases are enriched at the top 5% quantile for all cell types, increasing risk by 3- to 5-fold. Individuals in the tail of monocyte-specific scores are at higher risk compared to other groups. The difference in the OR scale between UKBB2 and EPIC should be noted. Data points in each panel are odds ratios estimated from a logistic regression and error bars represent standard errors on the values.

monocytes ($R^2 = 3.1\%$, $P = 4$e-16) and B cells ($R^2 = 2\%$, $P = 2.3$e-11) as summarized in Table 1.

Grouping individuals according to their CPRS indicated that subjects within the top 5% of all CPRS scores were at 3- to 5-fold increased risk for MS relative to those in the median quantile. Remarkably, subjects with higher monocyte-specific scores were at higher risk compared to other groups (Fig. 4a–c and Supplementary Data 7). Similar patterns were observed in the EPIC dataset, however, the increase of risk in tails of CPRS distributions is greater than UKBB2 (Fig. 4d–f and Supplementary Data 7).

**Association of CPRS with MS phenotypes**. The MRI-based phenotypic outcomes are hallmarks of disease activity capturing regional tissue loss in the CNS. We assessed the association of CPRS based on both all and unique cell-specific variants with the baseline values of MS neuroimaging markers, including volumetric measurements of the brain (BV), white matter (WMV), gray matter

(GMV), and cerebrospinal fluid (CSF) in the MS subjects of the UCSF-EPIC cohort ($n = 461$). Associations of CPRS scores with MS phenotypes were examined in linear regression models corrected for age at examination, gender, and disease duration, and goodness of fit was measured by $R^2$. Associations remained significant with WMV after correcting for multiple testing ($P < 0.05$). The highest observed association was between monocyte-specific scores and WMV ($\beta = -0.13$, $R^2 = 2.1\%$) as shown in Supplementary Table 4. Excluding *MHC* further improved associations of monocyte-specific scores with both WMV ($\beta = -0.14$, $R^2 = 2.2\%$, $P < 0.05$) and BV ($\beta = -0.10$, $R^2 = 1.66\%$, $P < 0.05$) as summarized in Supplementary Table 5. A relatively strong association of WMV with CPRS incorporating only unique SNPs in microglia ($\beta = -0.13$. $R^2 = 2.03\%$, $P < 0.05$) and monocytes ($\beta = -0.11$, $R^2 = 1.54\%$, $P < 0.05$) was observed (Table 2). Lastly, weak positive associations were found between the CSF volume and CPRS of unique SNPs in B cell ($\beta = 0.08$, $R^2 = 0.85\%$, $P < 0.05$) and monocyte ($\beta = 0.08$, $R^2 = 0.92\%$, $P < 0.05$).

**Table 2 Phenotype association of CPRS based on unique SNPs.**

| Phenotype | Combined | | B cell | | Monocyte | | Microglia | |
|---|---|---|---|---|---|---|---|---|
| | R$^2$ (%) | β | R$^2$ (%) | β | R$^2$ (%) | β | R$^2$ (%) | β |
| BV | 1 | −0.08 | 1.03 | −0.08 | 1.08 | −0.08 | 1.32 | −0.09 |
| WMV | 1.65[a] | −0.12 | 0.99[a] | −0.09 | 1.54[a] | −0.11 | 2.03[a] | −0.13 |
| GMV | 0.18 | −0.03 | 0.44 | −0.05 | 0.28 | −0.04 | 0.28 | −0.04 |
| CSF | 0.82 | 0.07 | 0.85[a] | 0.08 | 0.92[a] | 0.08 | 0.82 | 0.08 |

[a]$P < 0.05$.

Replicating these associations in the limited number of MS cases in the UKBB2 with available phenotypic information ($n = 89$) was not successful (Supplementary Table 6). This is likely due to the small cohort size, differences in measurement protocols, and treatment trajectories. Lastly, none of the CPRS-phenotype associations in the UK Biobank non-MS subjects with MRI data ($n = 819$) turned out to be significant (Supplementary Table 7).

The association of CPRS with disease activity was also investigated in the UCSF-EPIC cohort. Participants ($n = 464$) were grouped according to whether they had experienced one or more relapses within a 5-year interval from the baseline visit, regardless of disease worsening within the same timeframe. An increase in both B cell- and microglia-specific risk was associated with relapse activity ($\beta_{\text{B-cell}} = 0.29$, $P < 0.005$; $\beta_{\text{microglia}} = 0.29$, $P < 0.005$; $\beta_{\text{monocyte}} = 0.25$, $P < 0.05$), which remained significant after correcting for age, sex, and disease duration.

## Discussion

In this study, we leveraged updated algorithms and GWAS summary statistics, together with regulatory annotations (DNase-seq, ATAC-seq, scATAC-seq, ChIP-seq) and chromatin interaction maps (PCHiC, HiChIP), to identify causal cell types and genes in MS. We document the significant enrichment of MS-risk associations in OCRs of microglia and peripheral immune cell types, the later driven by active enhancers in B cells and monocytes. The lack of scATAC-seq data from patients with MS is a limitation of this analysis. Future studies will be required to generate such datasets. Joint analysis on scATAC-seq data from healthy individuals and MS patients will help us better understand MS pathogenesis.

The importance of B cells in MS has been confirmed by the success of B cell-depleting anti-CD20 therapies[25–27]. Our recent work demonstrates widespread hypomethylation in CD19 + B cells at clinical disease onset[28], posing a mechanistic link to the remarkable clinical efficiency of the anti-CD20 antibody treatments for this disease. Increasing attention has been given to the role of memory B cells in MS, which are enriched in the cerebrospinal fluid of MS patients[29–31], and indeed, a recent study concluded that the genetic enrichment in B cells is driven by OCRs in memory B cells[4]. Here, using single-cell and bulk chromatin accessibility data, we show that MS GWAS signals are significantly enriched in both naïve and memory B cells. These results indicate that both naïve B cells and memory B cells play important roles in MS susceptibility.

Using scATAC-seq data, we found MS GWAS signals were significantly enriched in microglia but not in other brain cell types. Using gene expression data, a previous study showed that MS risk genes are only significantly enriched in microglia within the CNS[1]. To our knowledge, our study demonstrates for the first time that MS GWAS signals are significantly enriched in regulatory regions of microglia, providing direct genetic evidence for microglia involvement in MS susceptibility.

Additionally, our analysis shows that MS GWAS signals are significantly enriched in active enhancers. It is widely accepted that the enhancers regulate target genes through 3D chromatin interactions. Therefore, we applied H-MAGMA framework to predict the putative causative genes associated with MS risk in B cells, monocytes, and microglia. By comparing different cell types, we further identified shared and cell type-specific genetic signatures. GO analysis on the shared genes is consistent with the previously established role of cytokine signaling, leukocyte activation, antigen processing and presentation, and NF-kappa B signaling[32–35]. In contrast, cell-type specific genes revealed the involvement of other pathways such as "G2/M checkpoints" and "PRC2 methylates histones and DNA", which are enriched in B cell- and microglia- unique genes, respectively. Consistent with our analysis, the predicted microglia-specific gene *DNMT3A* has been recently reported in a scRNA-seq study to be over-expressed in one cluster of microglial cells from MS patients (Fold increase: 1.21, $P = 8.37e-08$, FDR = 0.0028)[36]. An extended body of data is consistent with Epstein-Barr virus (EBV) infection triggering the development of MS[37]. Previous studies have validated that EBV infection of B cells results in epigenetic changes of both EBV and cellular genomes, including expression changes in DNA methyltransferases (DNMTs), and the following widespread expression changes in cellular genes[38,39]. Altogether, these findings suggest that DNMTs are involved in the early development of MS through the epigenetic control of immune cells, especially B cells and microglia.

Recent studies demonstrated that polygenetic risk scores are informative measures of risk in MS and other autoimmune diseases[40–43]. Cell-specific risk scores incorporating variants based on 3D chromatin interaction profiles also show statistically significant association with risk, suggesting that disease-associated variants mediate susceptibility to a large degree through chromatin interactions. Strong association of CPRS based on unique genetic markers in monocytes and B cells further confirm the importance of these cell types in MS susceptibility. Remarkably, CPRS based on unique SNPs in all three cell-types, particularly in microglia, have significant associations with WMV. Furthermore, the effect of *MHC* genetic burden inclusion had a different impact on phenotype association as compared to disease susceptibility, suggesting distinct mechanisms through which *MHC* regulates MS risk versus progression. To our knowledge, this is the first study directly linking cell-type specific risk burden statistics to MS phenotypes, although the clinical implications of these findings require further functional validation.

In summary, our analysis provides biological insights into MS genetic susceptibility and pinpoint at B cells and microglia as key mediators of disease risk variants. Further studies will be required to functionally validate the regulatory networks in a cell-specific manner.

## Methods

**GWAS summary statistics.** The International Multiple Sclerosis Genetics Consortium (IMSGC) GWAS summary statistics for 8,589,720 SNPs coming from 14,802 subjects with MS and 26,703 controls of European ancestry[1] were used for the analysis. We additionally obtained GWAS summary statistics from multiple autoimmune and psychiatric disorders: systemic lupus erythematosus (SLE)[12], rheumatoid arthritis (RA)[13], celiac disease (CD)[14], inflammatory bowel disease (IBD)[15], systemic sclerosis (SS)[16], type 1 diabetes (T1D)[17], Alzheimer disease (AD)[18], schizophrenia (SCZ)[19], and bipolar disorder (BPD)[20]. The sample sizes, ancestry information and covered SNPs numbers of these GWAS data were presented in Supplementary Data 8.

**Epigenetic datasets.** Single-cell ATAC-seq (scATAC-seq) peaks of peripheral blood and adult human brain were obtained from Corces et al.[7] and Chiou et al.[8]. For each dataset, the peaks that were uniquely presented in half of all cell types or less were defined as cell-type-specific peaks. Bulk DNase I–hypersensitive sites and ATAC-seq peak data were downloaded from ENCODE[44–46], Roadmap Epigenomics[47], and Blueprint projects[48] for available cell types or tissue from the immune system and CNS (Supplementary Data 9). The histone modification chromatin immunoprecipitation sequencing (ChIP-seq) peaks were downloaded from ENCODE[44–46], Roadmap Epigenomics[47], and Blueprint projects (Supplementary Data 9). The Candidate cis-Regulatory Elements (cCREs) defined by DNase hypersensitivity sites, histone modifications, and CTCF-binding data were obtained from ENCODE project[46].

**Enrichment of GWAS associations within regulatory annotations.** We applied GARFIELD v2 (GWAS Analysis of Regulatory or Functional Information Enrichment with LD correction)[5] to the MS GWAS discovery summary statistics to calculate the enrichment of GWASs associations within cell-type specific annotations (DNase I–hypersensitive sites, ATAC-seq peak, ChIP-seq data and cCREs), using the default LD information available from the package. Briefly, the LD was calculated using PLINK with --tag-r2 0.01 --tag-kb 500 (and --tag-r2 0.8 --tag-kb 500) flags to identify all proxies within a 1-Mb window around each variant, at $R^2$ 0.01 and 0.8. The data were computed from 3,621 European individuals from UK10K project. Variants were annotated in 0/1 format based on the overlap information with each regulatory annotation. For the MS GWAS enrichment analysis, we performed five independent GARFIELD analyses at eight GWAS $P$-value thresholds (T < 0.05 to T < $10^{-8}$) based on regulatory annotation obtained from difference resources: 1) cell-type-specific regulatory annotations derived from scATAC-seq data of peripheral blood and adult human brain. 2) open chromatin annotations in 424 cell lines and primary cell types available from GARFIELD package[5], 3) DNase-seq, ATAC-seq, and histone modification ChIP-seq data downloaded from ENCODE project for immune and brain cell types, 4) DNase-seq, ATAC-seq, and histone modification ChIP-seq data downloaded from Blueprint project for immune cell types, 5) cCREs from ENCODE project for immune cell types. GARFIELD was also run on the GWAS summary statistics from SLE, RA, CD, IBD, SS, T1D, AD, SCZ and BPD to calculate the enrichment of genetic associations of these diseases within cell-type-specific regulatory annotations derived from scATAC-seq data.

**H-MAGMA analysis.** To predict the MS risk genes associated with statistically significant risk SNPs we employed the H-MAGMA analysis[6] by incorporating chromatin interaction profiles in B cells, monocytes, and microglia. The GWAS summary statistics (14,802 subjects with MS and 26,703 controls) was used as one input file for the analysis. The promoter capture Hi-C (PCHiC) data[22] from B cell and monocytes and H3K4me3 HiChIP data from microglia[23] were used to generate the gene-SNP annotation files for the H-MAGMA analysis. First, the exonic and promoter SNPs were annotated to the genes based on the genomic location information from the reference of Gencode v26 (GRCh37). A promoter was defined as 2-kb upstream of the transcription start site (TSS) of each gene isoform. Next, the chromatin interaction regions were overlapped with Gencode v26 (GRCh37) exon and promoter coordinates to identify exon-based and promoter-based interactions. After that, the SNPs that interact with gene promoters or exons were annotated to the interacting genes. The chromatin interaction-based annotations were then combined with location-based annotations to generate the final gene-SNP annotation file for each cell type. The default setting of H-MAGMA v1.08 was used to run the pipeline. After generating the gene level $P$ values, the false discovery rate (FDR) values were calculated using the function p.adjust in R, restricting the risk genes to protein-coding genes with FDR < 0.05.

The risk genes predicted by H-MAGMA were compared between each cell type to identify the common and unique risk genes. To capture the biological functions associated with the gene lists, we applied Metascape[49], a web-based platform, to provide comprehensive gene annotation and enrichment analysis.

### Genotype-phenotype datasets

*UK Biobank.* The UK Biobank prospective cohort is an open resource providing genetic, phenotypic, and several health-related indicators for over 500,000 individuals residing in the United Kingdom[50]. Genome-wide genotype data have been collected for all participants; details described by Bycroft et al.[50] Prior to polygenic risk score validation and testing, bi-allelic variants were filtered for low imputation quality (INFO < 0.6), low minor allele frequencies (< 1%), genotype missingness (> 10%), and deviating from the Hardy-Weinberg equilibrium (P < 1e-6). Individuals with 'British' ethnicity and categorized as Caucasian according to genetic principal components were kept for further analyses. Non-MS individuals with no record of self-reported autoimmune, neurodegenerative, and mononucleosis infection diagnoses according to ICD-coded diagnosis were included. Related individuals were excluded according to their kinship coefficient (> 0.0844). Individuals withdrawn from the informed consent, and with low genotype quality (discordance between genetically reported and inferred sex and putative sex chromosome aneuploidy) were removed. Age of MS diagnosis is self-reported. The total number of cases and controls were 601 and 109,990 in the UK Biobank phase 1, and 1354 and 252,065 in the UK Biobank phase 2, respectively. Selected imaging-derived phenotypes normalized to the head size of 89 MS cases produced by an image-processing pipeline developed on behalf of UK Biobank[51,52], was used to phenotype association.

*UCSF EPIC.* Genetic and baseline clinical and imaging data of the UCSF-EPIC cohort including 462 MS patients were utilized for studying the association between polygenic risk scores and selected MRI and clinical phenotypes[53]. Variant- and individual-level quality controls were performed in accordance with the UK Biobank. Clinical and MRI outcomes of EPIC patients included volumetric measurements of the total brain (BV), total grey matter (GMV), peripheral gray matter (pGMV), white matter (WM), and cerebrospinal fluid (CSF). All studies were approved by the UCSF Institutional Review Board, and all datasets involving human samples acquired informed consent from participants.

**Cell-specific polygenic risk score (CPRS) and phenotype association.** We used PRSice-2[54] to generate and validate polygenic risk score (PRS) for MS and non-MS subjects in the UK Biobank phase 1 cohort (UKBB1). Significance values and effect sizes from the IMSGC summary statistics were utilized to obtain the best CPRS model, which was then tested in the UK Biobank phase 2 (UKBB2) and UCSF-EPIC target cohorts. CPRS were computed for SNPs annotated to cell-specific gene sets identified by the H-MAGMA algorithm, the gene-SNP annotations for each cell type were based on chromatin interaction and location as indicated in H-MAGMA analysis. The CPRS for each subject is generated by taking the weighted sum of the pruned effect alleles. The $r^2$ of 0.1 was used to ensure the inclusion of only independent effects. Scores were optimized across a range of $P$ value thresholds (5e-08, 5e-07, 5e-06, 5e-05, 0.0005, 0.005, 0.05, 0.1, 0.2, 0.3, 0.4, 0.5, and 1) in the base dataset. Coefficient of determination adjusted for MS prevalence in the white population and area under the curve (AUC) was used as measures of prediction accuracy.

Regression models were used to assess phenotype association of CPRS. Covariates included in the phenotype association test were age at examination, gender, and disease duration.

**Statistics and reproducibility.** The statistical tests used in this study were performed using R v3.6, and details statistical analyses were described within the methods section. Significantly GO terms and KEGG pathways were defined under the threshold of $P$ value < 0.05. The significant $P$ values were marked with *$P$ < 0.05, **$P$ < 0.01.

**Reporting summary.** Further information on research design is available in the Nature Portfolio Reporting Summary linked to this article.

## Data availability

The source data underlying Fig. 3b, c are presented in Supplementary Data 4 & 5. The source data underlying Fig. 4 are presented in Supplementary Data 7. The source studies for the summary statistics are presented in Supplementary Data 8. Links for publicly available epigenetic datasets are included in Supplementary Data 9. The cCREs data were downloaded from https://screen.wenglab.org/. Any other data that support the findings of this study are available through application to UK Biobank or request from the corresponding author.

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

## Acknowledgements

This study was supported by grants from the National Multiple Sclerosis Society (RFA-2104-37474 to J.R.O.), the National Institutes of Health (R35NS111644 to S.L.H.) and the Valhalla Foundation (to S.L.H.). Q.M. is supported by a postdoctoral fellowship from the National Multiple Sclerosis Society (FG-2108-38348). H.S. was supported by a postdoctoral fellowship from the National Multiple Sclerosis Society (FG-1807-31603). We acknowledge the International Multiple Sclerosis Genetics Consortium for access to GWAS summary statistics, and the UK Bionbank (Project ID: 59309) and University of California San Francisco MS-EPIC Study team for access to phenotypic data.

## Author contributions

Q.M., H.S., and J.R.O. conceived and supervised the study. Q.M., H.S., and J.R.O. analyzed the data. Q.M., H.S., J.R.O., A.D., S.E.B., B.A.C.C., S.L.H. and R.G.H. wrote the paper. B.A.C.C., and S.L.H. supervised sample acquisitions and linked clinical data. R.G.H. supervised MRI data acquisition, images processing, and quality control. All authors read and approved the final manuscript.

## Competing interests

S.L.H. currently serves on the scientific advisory board of Accure, Alector, Annexon, board of directors of Neurona, and has previously consulted for BD, Moderna, and NGM Bio. S.L.H. also has received travel reimbursement and writing support from F. Hoffmann-La Roche and Novartis AG for anti-CD20-theapy-related meetings and presentations. Other authors declare no competing interests.

**Additional information**

