## [Peer Review File · Communications Biology]

Reviewers' comments:

Reviewer #1 (Remarks to the Author):

Summary of manuscript

In this work Ma and colleagues seek to elucidate the causative genes and cell types underlying multiple sclerosis (MS) development by integrating MS GWAS signals with publicly available datasets from immune and adult CNS cell types. They apply two established methods, GARFIELD and H-MAGMA, to identify the enrichment of MS genetic risk associations in regulatory annotations at the cellular level (using GARFIELD) and to integrate 3D Hi-C in three specific cell types (B cells, monocytes and microglia) to identify cell type specific MS risk genes (using H-MAGMA). This ultimately led to the authors constructing cell specific polygenic risk-scores from these three cell types that they use to associate clinical phenotypes relevant to MS in both the UK Biobank and the UCSF-EPIC cohort (494 neurologist-diagnosed MS cases and 449 controls). Together these findings contribute to the understanding of the specific cell types that MS risk variants identified from GWAS impact, thus potentially indicating the target cell types for disease treatment.

Overall impression of the work:

This work is an extremely thorough analysis that refines and hones the seminal work of the IMSGC in their 2019 MS GWAS. The analyses and their corresponding methodologies are detailed enough to enable replication of this work using the same datasets, or other relevant datasets as they become available. The statistical analyses applied are appropriate. This work will be of high interest to the MS field, particularly to neurobiologists integrating their understanding of MS genetic risk with disease biology.

Specific comments with recommendations addressing each comment:

1) Prior findings detailed in the H-MAGMA manuscript (Set et al. 2020, Nature Neuroscience) do support the involvement of non-immune cell types, and processes as enriched, in MS. Particularly oligodendrocyte processes (myelination), and neurovascular cell types (endothelial cells). I note that Corces et al. which is the adult CNS cell type scATAC-seq dataset used in the current work does not contain a neurovascular population. Does this together imply that a limitation of this work is in the datasets applied? And that non-immune based mechanisms may have been missed?

2) The authors report an association between cell specific polygenic risk scores and MRI-based MS relevant phenotypes in the 461 people with MS in the UCSF-EPIC cohort. Given that there are control subjects within this cohort, who presumably have the same MRI-based phenotypic data, could the authors consider extending the CPRS analysis to controls, or controls and people with MS in combination? It would be interesting to identify any consistent directional effects for the CPRS associations in individuals without MS.

Minor comments:

Fig S1. Spacing needed between disease and critical in 'disease critical cell types' box.
Figure 1 A. Oligodendrocyte label is truncated on radial plot.

Reviewer #2 (Remarks to the Author):

The aim described in this report was to identify using more recently described algorithms the physiological processes associated with the development of MS. GWAS studies have revealed a large number of genetic variants that are linked to MS risk which reside in noncoding regions of the genome that are not easily equated to any functional change that can be unequivocally associated with a disease mechanism.

The authors used two key software packages, GARFIELD and H-MAGMA, both of which show promise in revealing robust associations with the functional consequences of genetic change occurring in noncoding regions of the genome. The data presented in the manuscript is consistent with other datasets available (ENCODE and the Blueprint projects) thereby increasing the reliability of GARFIELD and its ability to identify open chromatin.

The findings of this study are important and reveal at a cellular level epigenetic marks that are associated with MS aetiology. It would be useful to provide at least some evidence that these results are specific to MS and not associated with other autoimmune disorders such as inflammatory bowel disease (often co-existing with MS). This is especially important if DNMT3A is considered as a significant finding.

Several questions remain about this gene and its role in MS progression and or initiation. Given the potential role of EBV in the etiology of MS some discussion about how it might influence epigenetic control in B-cells and monocytes would add to the value of this manuscript.

Minor points

In the description of the UK biobank there is a statement with "he" in it - this appears to me to be a typo.

The frequency of MS in the UK biobank seems quite high, compared to a recent survey by Walton et al. 2020. Some explanation should be provided to explain this discrepancy.

Reviewer #3 (Remarks to the Author):

In this work, the authors conducted systematic mapping for multiple sclerosis (MS) GWAS signals to their cell-type context. Overall, the GWAS signals are enriched in B cell and monocyte/microglial cell-types, consistent with the current known disease mechanism and therapeutic pathways. Here are a few comments.

Major:

1. scATAC data is obtained for the Alzheimer's and Parkinson's diseases, which might not be proper to reflect the MS context. It is not clear whether the authors use the control samples only or not. If the scATAC MS dataset is not available, can the authors at least include a limitation paragraph in the discussion part.
2. I suggest the authors could provide a negative control (such as Neurons) for CPRS performance to justify that those signals are specifically disease-related cell types.
3. The metric coefficient of determination (R2) is not well-defined. Can the authors add more description in methods part for the regression model used in CPRS and MS phenotypes (such as what covariates included).

Minor:

1. Can the authors share the genes list enriched in B cell, Monocyte, and Microglia? They maybe useful for other researchers to continue work on these prioritized genes.
2. In discussion, I suggest the authors could add one paragraph to connect the current findings. The current results seem separated from each other. Therefore, the story of the manuscript is not well-interpreted and impressive despite the huge amount of work here.
3. Table for R2, should add % in the column names.
4. This sentence is confusing. "Specifically, excluding MHC burden decreased AUC by 3 to 5% and R2 by 0.2 to 1.5%." Please revise.

Dear Dr. Bahlo and Dr. Inglis,

We are pleased to submit a revised version of our manuscript titled “Identifying multiple sclerosis disease-critical cell types and genes by integration of epigenetic and genetic profiles”. We appreciate the constructive comments of the reviewers, based on which we have made revisions that have undoubtedly improved the manuscript. All coauthors have approved the revised manuscript. Below is a point-by-point response to the reviewers’ comments. Modifications in the revised manuscript are highlighted in yellow, and the significant body of new data is presented in the revised **Figure S3 and Table S12**.

Reviewer #1:

Summary of manuscript

In this work Ma and colleagues seek to elucidate the causative genes and cell types underlying multiple sclerosis (MS) development by integrating MS GWAS signals with publicly available datasets from immune and adult CNS cell types. They apply two established methods, GARFIELD and H-MAGMA, to identify the enrichment of MS genetic risk associations in regulatory annotations at the cellular level (using GARFIELD) and to integrate 3D Hi-C in three specific cell types (B cells, monocytes and microglia) to identify cell type specific MS risk genes (using H-MAGMA). This ultimately led to the authors constructing cell specific polygenic risk-scores from these three cell types that they use to associate clinical phenotypes relevant to MS in both the UK Biobank and the UCSF-EPIC cohort (494 neurologist-diagnosed MS cases and 449 controls). Together these findings contribute to the understanding of the specific cell types that MS risk variants identified from GWAS impact, thus potentially indicating the target cell types for disease treatment.

Overall impression of the work:

This work is an extremely thorough analysis that refines and hones the seminal work of the IMSGC in their 2019 MS GWAS. The analyses and their corresponding methodologies are detailed enough to enable replication of this work using the same datasets, or other relevant datasets as they become available. The statistical analyses applied are appropriate. This work will be of high interest to the MS field, particularly to neurobiologists integrating their understanding of MS genetic risk with disease biology.

Response: We thank the reviewer for the positive comments. Please find our responses to specific comments below:

1) Prior findings detailed in the H-MAGMA manuscript (Set et al. 2020, Nature Neuroscience) do support the involvement of non-immune cell types, and processes as enriched, in MS. Particularly oligodendrocyte processes (myelination), and neurovascular cell types (endothelial cells). I note that Corces et al. which is the adult CNS cell type scATAC-seq dataset used in the current work does not contain a neurovascular population. Does this together imply that a limitation of this work is in the datasets applied? And that non-immune based mechanisms may have been missed?

Response: We thank the reviewer for highlighting this important concern. We extended the analysis using regulatory annotations denoting open chromatin regions (OCRs) in 424 cell lines or primary cell types and performed follow-up analyses in the immune system and CNS tissue available from the Encyclopedia of DNA Elements (ENCODE) and the Blueprint projects (**Fig. 1B-C**). In this extended

analysis, immune cell types remain the principal target of GWAS associations enrichment, especially B cells and monocytes (**Fig. 1C and Fig. S2**), whereas modest enrichment (odds ratio values range from 1.34 to 2.52 at GWAS P-value threshold $T < 10^{-5}$) was observed in some CNS chromatin accessibility datasets, including brain microvascular endothelial cells. (**Fig. 1C**).

2) The authors report an association between cell specific polygenic risk scores and MRI-based MS relevant phenotypes in the 461 people with MS in the UCSF-EPIC cohort. Given that there are control subjects within this cohort, who presumably have the same MRI-based phenotypic data, could the authors consider extending the CPRS analysis to controls, or controls and people with MS in combination? It would be interesting to identify any consistent directional effects for the CPRS associations in individuals without MS.

Response: We appreciate the reviewer’s important comment. Unfortunately, the healthy controls in the UCSF-EPIC cohorts lack MRI data, but we tested the CPRS-phenotype association in UK Biobank non-MS individuals and added the following to page 10 of the main text:

“Lastly, none of the CPRS-phenotype associations in the UK Biobank non-MS subjects with MRI data (n = 819) turned out to be significant (**Table S12**).”

Table S12. Phenotype association of CPRS of non-MS subjects in UK Biobank

Phenotype	Combined		B cell		Monocyte		Microglia	
	R ² (%)	β	R ² (%)	β	R ² (%)	β	R ² (%)	β
BV	3.55	1	4.11	0.92	2.19	0.99	0	0.21
WMV	4.11	1.6	4.77	1.97	2.54	1.06	0	0.8
GMV	3.35	0.19	3.87	-0.24	2.06	0.62	3	-0.34
CSF	0.1	2.82	0.1	3.71	2.38	0.2	0.1	3.51

Minor comments:

Fig S1. Spacing needed between disease and critical in ‘disease critical cell types’ box.

Figure 1 A. Oligodendrocyte label is truncated on radial plot.

Response: We thank the reviewer. All typos have been corrected in the revised manuscript.

Reviewer #2:

The aim described in this report was to identify using more recently described algorithms the physiological processes associated with the development of MS. GWAS studies have revealed a large number of genetic variants that are linked to MS risk which reside in noncoding regions of the genome that are not easily equated to any functional change that can be unequivocally associated with a disease mechanism.

The authors used two key software packages, GARFIELD and H-MAGMA, both of which show promise in revealing robust associations with the functional consequences of genetic change occurring in noncoding regions of the genome. The data presented in the manuscript is consistent with other datasets available (ENCODE and the Blueprint projects) thereby increasing the reliability of GARFIELD and its ability to identify open chromatin.

The findings of this study are important and reveal at a cellular level epigenetic marks that are associated with MS aetiology. It would be useful to provide at least some evidence that these results are specific to MS and not associated with other autoimmune disorders such as inflammatory bowel disease (often co-existing with MS). This is especially important if DNMT3A is considered as a significant finding.

Several questions remain about this gene and its role in MS progression and or initiation. Given the potential role of EBV in the etiology of MS some discussion about how it might influence epigenetic control in B-cells and monocytes would add to the value of this manuscript.

Response: We thank the reviewer for the favorable comments on our study. Following the reviewer's suggestion, we tested the heritability enrichment using scATAC-seq datasets for GWAS associations of several immune-related and psychiatric disorders: systemic lupus erythematosus (SLE), rheumatoid arthritis (RA), celiac disease (CD), inflammatory bowel disease (IBD), systemic sclerosis (SS), type 1 diabetes (T1D), Alzheimer disease (AD), schizophrenia (SCZ) and bipolar disorder (BPD). As shown in the **revised Figure S3**, B cells, microglia and monocytes showed the highest enrichment in MS compared to other immune-related and psychiatric disorders. The new data is added to page 5 of the revised manuscript.

Furthermore, in the revised version, we have modified the discussions about DNMT3A on page 12 of the main text as following:

"In contrast, cell-type specific genes revealed the involvement of novel pathways such as "G2/M checkpoints" and "PRC2 methylates histones and DNA", which are enriched in B cell- and microglia-unique genes, respectively. Consistent with our analysis, the predicted microglia-specific gene DNMT3A has been recently reported in a scRNA-seq study to be over-expressed in one cluster of microglial cells from MS patients (Fold increase: 1.21, P= 8.37E-08, FDR=0.0028). An extended body of data is consistent with Epstein-Barr virus (EBV) infection triggering the development of MS. Previous studies have validated that EBV infection of B cells results in epigenetic changes of both EBV and cellular genomes, including expression changes in DNA methyltransferases (DNMTs), and the following widespread expression changes in cellular genes. Altogether, these findings suggest that DNMTs are involved in the early development of MS through the epigenetic control of immune cells, especially B cells and microglia."

Minor points

In the description of the UK biobank there is a statement with "he" in it - this appears to me to be a typo. The frequency of MS in the UK biobank seems quite high, compared to a recent survey by Walton et al. 2020. Some explanation should be provided to explain this discrepancy.

Response: We agree with the reviewer that the frequency of cases in UK Biobank is higher than the MS prevalence in the white population, which is why we corrected the associations reported in this study for MS prevalence, as explained on page 7 of the revised manuscript as follows:

"The predictive power of CPRS for MS risk is very similar across different cell subtypes as well as the combined score in all datasets, indicated by the coefficient of determination (R^2) corrected for disease

prevalence in the white population in Western Europe (0.00127) as well as the area under the curve (AUC) as shown in **Table S5**.” It is also mentioned in the caption of **Table 1** on page 8. The recent survey by Walton et al. showed that the prevalence per 100,000 population in European is 142.81 [142.53, 143.08], which is slightly higher than the reference used in this study. All typos have been corrected in the revised manuscript.

Reviewer #3:

In this work, the authors conducted systematic mapping for multiple sclerosis (MS) GWAS signals to their cell-type context. Overall, the GWAS signals are enriched in B cell and monocyte/microglial cell-types, consistent with the current known disease mechanism and therapeutic pathways. Here are a few comments.

Major:

1. scATAC data is obtained for the Alzheimer’s and Parkinson’s diseases, which might not be proper to reflect the MS context. It is not clear whether the authors use the control samples only or not. If the scATAC MS dataset is not available, can the authors at least include a limitation paragraph in the discussion part.

Response: We sincerely apologize for the lack of clarity. Corces et al. used a cohort of cognitively healthy individuals to generate the scATAC-seq data. On page 4 of the revised manuscript, we have modified the text as follows: “Two studies have generated accessible chromatin reference maps from single-cell ATAC-seq (scATAC-seq) screenings on healthy peripheral blood and brain tissue from cognitively healthy individuals”.

We have also added a paragraph describing the limitation of the scATAC-seq dataset used in this study on page 11 of the revised discussion:

“The lack of scATAC-seq data from patients with MS is a limitation of this analysis. Future studies will be required to generate such datasets. Joint analysis on scATAC-seq data from both healthy individuals and MS patients will help us better understand MS pathogenesis”.

2. I suggest the authors could provide a negative control (such as Neurons) for CPRS performance to justify that those signals are specifically disease-related cell types.

Response: We thank the reviewer for the suggestion. Based on the first reviewer concern on the involvement of non-immune cell types/processes in CNS, and our following analysis using bulk chromatin accessibility data showing modest enrichment in chromatin accessibility in some CNS datasets (please see our response on page 1 of this letter), lung tissue instead of neurons, was used to develop a true CPRS negative-control. No significant association was observed between lung-CPRS and either MS risk or neuroimaging phenotypes (shown Table 1 and 2 on the next page), which further supports the specificity of selected disease-relevant cell types in this study. Nevertheless, we would like to emphasize that CPRS calculation was particularly intended to confirm the significance of disease-critical cell types and genes identified using GARFIELD and H-MAGMA. The intricate chromatin interactions modulating long-range SNP effects, which are typically ignored in classical post-GWAS prioritization approaches (such as MAGMA), may result in complex gene-SNP associations impacting the determination of cell-specificity. Since the precision of cumulative scores is affected by such non-specific SNP effects, chromatin accessibility and H-MAGMA analyses were utilized in our study design for identifying disease-relevant cell types prior to CPRS calculation.

Table 1. Prediction accuracy of CPRS with unique SNPs

Model ($r^2 = 0.1$)	UKBB2 1354 cases/ 252,065 controls			UCSF-EPIC 494 cases/ 449 controls		
	R ^{2*} (%)	P	AUC (%)	R ^{2*} (%)	P	AUC (%)
B cell	3.3	7e-119	66.8	4.2	2e-20	70
Monocyte	3.2	3e-116	66	4.5	2e-21	70.7
Microglia	3.1	1e-114	66.5	4.3	9e-21	70.5
Lung	0.1	2e-5	52	0.3	0.003	55

Table 2. Phenotype association of lung-specific CPRS in UKBB2 and UCSF-EPIC

Model ($r^2 = 0.1$)	UKBB2 1354 cases/ 252,065 controls		UCSF-EPIC 494 cases/ 449 controls	
	R ² (%)	β	R ² (%)	β
BV	0.2	-2	0.6	-0.06
WMV	0.1	1.6	0	0
GM	0.9	-4.2	1.2	-0.08
CSF	0.6	4.1	0.1	0.03

*Adjusted for MS prevalence of 0.00127

3. The metric coefficient of determination (R²) is not well-defined. Can the authors add more description in methods part for the regression model used in CPRS and MS phenotypes (such as what covariates included).

Response: We added the following text on page 9 of the revised manuscript:

“Associations of CPRS scores with MS phenotypes were examined in linear regression models corrected for age at examination, gender, and disease duration, and goodness of fit was measured by coefficient of determination (R²).”

Minor:

1. Can the authors share the genes list enriched in B cell, Monocyte, and Microglia? They maybe useful for other researchers to continue work on these prioritized genes.

Response: The gene lists are included in **Table S2**.

2. In discussion, I suggest the authors could add one paragraph to connect the current findings. The current results seem separated from each other. Therefore, the story of the manuscript is not well-interpreted and impressive despite the huge amount of work here.

Response: Following the reviewer's suggestion, we have added and highlighted some discussions on page 11 of the revised manuscript to connect the current findings in the revised manuscript:
"Using scATAC-seq data, we found MS GWAS signals were significantly enriched in microglia but not in other brain cell types. Using gene expression data, a previous study showed that MS risk genes are only significantly enriched in microglia within the CNS. To our knowledge, our study demonstrates for the first time that MS GWAS signals are significantly enriched in regulatory regions of microglia, providing direct genetic evidence for microglia involvement in MS susceptibility."

3. Table for R2, should add % in the column names.

Response: We thank the reviewer. We have added % in the column names in the revised manuscript.

4. This sentence is confusing. "Specifically, excluding MHC burden decreased AUC by 3 to 5% and R2 by 0.2 to 1.5%." Please revise.

Response: We have revised the text on page 8 of the revised manuscript as follows:
"Specifically, excluding the MHC burden from the cumulative CPRS decreased the AUC values shown in Table S5 by 3% to 5% across all cell types."

REVIEWERS' COMMENTS:

Reviewer #1 (Remarks to the Author):

I have reviewed the authors responses to the reviewers comments and consider all my points addressed.

Reviewer #2 (Remarks to the Author):

The authors have addressed the concerns I raised in my original review. In addition, concerns raised by others appear to me to be adequately addressed.

Reviewer #3 (Remarks to the Author):

The authors have addressed all my concerns. I have no further comments.